# Predictive Models for Necrotizing Soft Tissue Infections: Are the Available Scores Trustable?

**DOI:** 10.3390/jcm14134550

**Published:** 2025-06-26

**Authors:** Sophie Tran, Kerry J. Pullano, Sharon Henry, Marcelo A. F. Ribeiro

**Affiliations:** 1School of Medicine, University of Maryland, Baltimore, MD 21201, USA; stran@som.umaryland.edu (S.T.); kerry.pullano@som.umaryland.edu (K.J.P.); 2R Adams Cowley Shock Trauma Center, University of Maryland, Baltimore, MD 21201, USA; sharon.henry@som.umaryland.edu

**Keywords:** necrotizing soft tissue infections (NSTI), Laboratory Risk Indicator for Necrotizing Fasciitis Score (LRINEC Score), neutrophil-to-lymphocyte ratio (NLR), platelet-to-lymphocyte ratio (PLR), NECROSIS score, POTTER score, in-hospital mortality

## Abstract

**Background:** Necrotizing soft tissue infections (NSTIs) remain a significant source of in-hospital morbidity and mortality in the U.S. and around the world, yet the need for a reliable tool to assess prognosis early in treatment remains unaddressed in the current medical literature. Many scoring systems have been developed; however, none have proven to be entirely reliable for use in patients with NSTIs. **Methods:** Using collected data through a PubMed and Google Scholar search, this review provides an overview of five scoring systems—LRINEC, platelet-to-lymphocyte ratio (PLR), neutrophil-to-lymphocyte ratio (NLR), NECROSIS, and POTTER—while highlighting potential areas for further improvement of these scoring systems or the conception of a novel, more effective system. **Results:** The most widely used scoring tool, the Laboratory Risk Indicator for Necrotizing Fasciitis Score (LRINEC), lacks high sensitivity and requires supplementation of other clinical parameters. The NECROSIS score offers a potentially improved system, though it lacks necessary external validation. NLR and PLR provide reliable measurements for immune response; however, they lack specificity for NSTI and require further research to determine parameters like cutoff values. The POTTER score, though not valid for use in patients with NSTI, poses a novel system utilizing AI technology and machine learning. **Conclusions:** This review concludes that further development of a reliable scoring system that accounts for the many factors involved in NSTI is required and may benefit from an integrative model like the POTTER score.

## 1. Introduction

Also known as the “flesh-eating disease”, necrotizing soft tissue infections (NSTIs) such as necrotizing fasciitis, necrotizing myositis, and necrotizing cellulitis are categorized as bacterial infections of the subcutaneous soft tissue surrounding muscles and nerves that can ultimately result in death depending on severity and timeliness of treatment [1]. NSTIs are most often caused by group A β-hemolytic streptococci, staphylococcal species, or both in combination following open wounds such as trauma or surgery [2]. Necrotizing soft tissue infections rapidly progress, and in their early stages often present with local edema, pain out of proportion, and erythema typically found in the lower and upper extremities, perineum, and trunk [3]. This clinical presentation can hold a similar differential to regular non-life-threatening cellulitis, which renders the disease difficult to diagnose [4].

As the disease advances, patients may exhibit dusky skin, gray-brown discharge or fluid-filled bullae, and subsequently cutaneous and deep fascial plane penetration. Late stages may lead to bacteremia and septic shock [5,6]. The incidence of NSTIs is increasing in the United States, with estimates ranging from 28,500 to 33,600 cases in 2018, or 10.3 and 8.7 per 100,000 persons, respectively [7]. Necrotizing fasciitis is most prevalent among patients with advanced age, immunosuppression, peripheral vascular disease, diabetes mellitus, obesity, recent surgery, or penetrating injury [3].

Over the past decades, researchers have worked towards establishing predictive models for diagnosing necrotizing fasciitis based on laboratory metrics in addition to physical examination, with the hope of decreasing the time between patient presentation and debridement. Significant morbidities are associated with debridement that is delayed more than 6–12 h after initial presentation, while significantly delayed debridement is associated with increased mortality [8]. *JAMA Surgery* identifies the current in-hospital mortality rate at 10–20% [1]. As such, early detection and accurate risk assessment are crucial keys to allow for timely intervention in these patients who have an elevated risk of morbidity and mortality.

A recent review published in *The Journal of Trauma and Acute Care Surgery* suggests that the best scoring system currently available for assessing a patient’s risk of NSTI is the Laboratory Risk Indicator for Necrotizing Fasciitis Score (LRINEC), which serves as a tool that can be used to increase suspicion for NSTI using laboratory findings such as white blood cell count (WBC), creatinine, sodium (Na), hemoglobin (Hgb), glucose, and C-reactive protein (CRP) [8]. However, even this system has limitations for diagnosis and does not provide an estimated mortality risk for each patient [8]. Several other scoring systems have been developed for diagnosis and mortality risk estimation utilizing various metrics. The aim of the present study is to provide an overview of five of the current scoring systems in use and identify directions for the further development or conception of an improved system.

## 2. Review of Current Scoring Systems

### 2.1. Relevant Sections

#### 2.1.1. LRINEC

The LRINEC was first introduced in 2004 through a study conducted at Changi General Hospital in Singapore as a comprehensive metric to determine the likelihood of developing necrotizing fasciitis. The score includes six variables: white cell count, CRP, Hgb, serum NA, glucose, and serum Cr [9]. The biochemical and hematologic variables were chosen based on laboratory changes typically seen in sepsis, which is strongly associated with necrotizing fasciitis, yet tangential to cellulitis or abscesses [9,10]. The numerical score is derived from the regression coefficient of each variable, which is then stratified to different risk levels: ≤5 low risk, ≥6 raise suspicion, ≥8 strongly predictive [9]. The original retrospective study found that the LRINEC significantly increased the early detection of the disease in a developmental cohort of 89 individuals treated for necrotizing fasciitis. While only 14.6% of these patients raised suspicion of necrotizing fasciitis based on clinical presentation, 89.9% of these same patients had a retrospective LRINEC ≥6, or a qualification for intermediate risk [9].

Since the release of this novel study in 2004, there have been multiple studies both confirming and challenging the sensitivity of the LRINEC. In 2022, a retrospective study including 70 patients treated for necrotizing fasciitis at a Level I trauma center in Germany found that the LRINEC did in fact decrease significantly after debridement (avg 0.663 points). However, the study also found that 37% of these patients with necrotizing fasciitis were incorrectly classified as low-risk based on the LRINEC. Further, the study concluded that the LRINEC should only be used in conjunction with other clinical parameters based on the high rates of false negatives [11].

A meta-analysis published in 2017 evaluating 16 studies (847 patients) confirmed a positive correlation between the LRINEC and true necrotizing fasciitis, with an average score of 6.06 for patients with necrotizing fasciitis and 2.45 for controls without the disease [12]. This study however also highlights the limitations of the LRINEC given its heavy reliance on hematologic values compared to a predictive system like APACHE II for pancreatitis, which takes other factors into account such as patient age, comorbidities, and clinical presentation [12,13].

#### 2.1.2. Neutrophil-to-Lymphocyte Ratio

The neutrophil-to-lymphocyte ratio (NLR) utilizes common lab values to demonstrate inflammation to chronic or acute injury [14]. The simple ratio between neutrophil and lymphocyte counts measured in peripheral blood allows for a relative comparison of the activities of a patient’s innate immune response and adaptive immune response at the time of lab draw [15]. An isolated elevation in neutrophil count and consequently elevated NLR suggests a condition has activated a patient’s systemic immune response syndrome (SIRS), thus presenting with a proinflammatory state as evidenced by an elevated neutrophil count [15]. The NLR has been shown to be independently and significantly associated with an increased risk of all-cause mortality [15]. Given that the NLR typically elevates in less than 6 h after acute physiological stress, the NLR presents a strong option as an early marker of acute stress compared to other laboratory parameters that do not appear as early in the inflammatory response, such as white blood count or c-reactive protein [15].

The NLR has been shown to be effective in the settings of cancer, rheumatoid disease, and autoinflammatory disease; however, there has been little evidence to demonstrate its utility in the setting of necrotizing soft tissue infections [16]. One retrospective study conducted at the 3rd Hospital, Hebei Medical University in China, in 2023, sought to determine its effectiveness in determining prognosis for patients with necrotizing fasciitis. The results revealed that an NLR ≥ 11.1 was independently associated with an increased risk of mortality after adjustment [14]. Similarly, in a single-center retrospective study including patients with Fournier’s Gangrene at a tertiary care hospital in Indonesia, the results show subjects with an NLR > 8 had a 12.062 times higher mortality risk than those with an NLR < 8 [16].

Though the mechanism is not fully understood, researchers suggest that the association of NLR to mortality is attributed to an inflammatory process that results in clinical findings such as neutrophilia and thrombocytosis, which can be quantified by a complete blood count (CBC) lab [14,17]. Given that CBCs are commonly drawn in an acute setting, the NLR remains a strong option as a useful biomarker in determining prognosis. However, a significant limitation includes the lack of fixed cutoff values for the NLR, which makes it difficult for providers to interpret these values in equivocal settings. Additionally, confounding variables such as obesity, HIV, and cancer must be considered [15].

#### 2.1.3. Platelet-to-Lymphocyte Ratio

In a similar manner to the NLR, the platelet-to-lymphocyte ratio (PLR) utilizes common laboratory values to demonstrate inflammatory processes related to acute and chronic injury. In the same retrospective study conducted at the 3rd Hospital, Hebei Medical University, investigating the utility of the NLR, the findings also demonstrated that a PLR ≥ 196.0 was independently associated with an increased risk of mortality after adjustment [14]. However, other studies have found that the PLR is not a useful predictor of morbidity and mortality [17]. There is very little data in the current literature investigating PLR pertaining to NSTIs. The two studies discussed here are the only two found at the time of this literature review. Their methods are both retrospective in nature; however, their contradictory results may be due in part to differences in sample sizes. Nonetheless, these conflicting results and the lack of consensus throughout the current literature suggest significant further research is necessary to determine the utility of the PLR in the prognostication of patients with NSTI. Given that a neutrophil count, platelet count, and lymphocyte count are metrics commonly available in a CBC lab draw requiring no extra cost and the calculations are simple, NLR and PLR have the potential to serve as useful tools for urgent care and primary care providers to identify possible high-risk patients that may require referral to higher level care centers [15].

#### 2.1.4. NECROSIS Score

The NECROSIS score is a more recent predictive model for NSTIs that combines patient vital signs, clinical presentation, and laboratory findings to determine the onset of necrosis. The final scoring system was derived after determining the most significant predictors among an extensive list of independent variables, including patient demographic (age, sex, weight, etc.) and radiographic findings, using multivariable regression analysis. This prospective study was performed among emergent general surgery patients over 2.5 years across 16 medical centers. The final NECROSIS score was narrowed to systolic blood pressure (SBP), violaceous skin, and white blood cell (WBC) count. The study found that, among all vital signs, systolic blood pressure still differed the most between patients with and without necrotizing infection, with an unusually stable cutoff of 120 mmHg. The authors use this value to caution physicians to be wary of potential NSTI even when patients appear hemodynamically stable, as the study found that only one in five NSTI patients met the criteria for septic shock upon initial presentation. Further, all continuous variables were represented categorically using cutoffs—systolic blood pressure (SBP) ≤ 120 mmHg and WBC ≥ 15 × 10^3^/uL—so that if a patient met the criteria for one of the variables they would be granted a score of 1 [18].

Using the derivation cohort, it was found that meeting the criteria for just one variable (score = 1) had higher sensitivity than scoring the maximum 3 points across all predictors, while a NECROSIS score of 3 had a 100% specificity. When comparing this new model to the LRINEC, the authors remark that the LRINEC would have missed 65% of NSTI patients using a cutoff of ≥6, whereas the NECROSIS score showed a sensitivity of 92% for patients with at least one NECROSIS variable [18].

Though this multicenter trial shows the NECROSIS score has great potential, the authors note that the study had several limitations. There was a large incidence of NSTIs across the centers involved in the study, which may limit the generalizability of its results to centers where NSTIs are less commonly encountered. Additionally, there was an increased availability of surgical expertise, access to ORs, and postoperative critical care resources that may have affected patient outcomes. Furthermore, the score has yet to be externally validated since this original derivation and validation trial. Pending the external validation of this predictive model, the researchers suggest that the combinative use of the NECROSIS score alongside the LRINEC may have great potential for improving patient outcomes overall [18].

#### 2.1.5. POTTER Score

The Predictive OpTimal Trees in Emergency Surgery Risk (POTTER) calculator presents a novel tool utilized by providers to determine 30-day outcomes in patients undergoing emergency operations [19]. The interactive calculator utilizes a machine learning-based algorithm, an artificial intelligence (AI) application in which machines recognize patterns based on their experiences from a given dataset, which allows for nonlinear variability as well as higher accuracy and interpretability than classical decision tree methods [20]. Essentially, the use of the machine learning method called Optimal Classification Trees (OCTs) allows the decision tree to adapt to each input and re-calculate risk after each variable. Further, the variables evaluated at each level of the tree are not the same, meaning the questions asked at each level change depending on the response to the previous question. This function is what allows for the nonlinear interactions between the variables and ultimately increases the accuracy of the tree, in contrast to the fixed interactions utilized in a classical logistic regression. Since its creation, the POTTER Score has been reformatted into a user-friendly iPhone and Android application which allows providers to determine accurate risk estimates at the bedside to better counsel patients and families prior to surgery [20].

Providers may choose to predict the 30-day postoperative mortality risk, overall postoperative mortality or any of the 18 specific postoperative complications identified in the ACS-NSQIP. The algorithm asks simple questions such as the age of the patient, laboratory and clinical values, medical history, and so on in an interpretable manner for the clinician, with each subsequent question depending on the previous answer. Although a strength of this decision tree-style format is that it allows the physician to follow along and interpret the questions being asked in a sequential manner, this type of format risks a significant limitation in that if it asks a question for which the provider does not have the answer readily available, there is no way for that provider to bypass that particular question. Thus, one is to assume the decision tree is rendered useless until that value or information is obtained and submitted to the calculator. Further, as with any AI program, the application is limited to the accuracy and comprehensiveness of the dataset upon which it was trained. Another strength, however, is that the POTTER score has the potential to be integrated into the electronic health record (EHR), which would offer even easier usability to clinical providers [20]. Although this type of predictive model has significant potential across many clinical fields, at this time, the tool has only been validated for use in patients undergoing emergency operations, particularly emergency general surgery [19,21].

A study performed at an urban academic medical center retrospectively assessed surgical debridement procedures involving a necrotizing skin and soft tissue infection (NSSTI) diagnosis between April 2015 and April 2020 and found that there were no significant regression equations with POTTER scores for the length of stay or the number of debridement operations. Thus, in that study involving 46 total cases, it was determined that the POTTER calculator was not a reliable predictor of 30-day postoperative all-cause morbidity in patients undergoing surgical debridement for NSSTI [22] (Table 1).

## 3. Discussion and Future Directions

Given the unreliability of the diagnostic and prognostic tools currently available, the general recommendation for surgical intervention in NSTI relies heavily on clinical suspicion alone, with available scoring systems like the LRINEC simply acting as adjuncts. Now, numerous studies—such as McDermott’s paper titled “Necrotizing Soft Tissue Infections–A Review”—are directly comparing the efficacy of different predictive scores to confirm utility and to guide the use of certain tools in the clinical field [1]. This review illustrates that novel predictive models for NSTIs have the potential to increase diagnostic power compared to the original LRINEC from 2004.

Early studies distinguished the limitations of the LRINEC and underlined the low sensitivity and high rates of false negatives among different patient samples. In fact, a newly published score titled J-LRINEC tests the validity among Japanese patients using all the same variables such as the LRINEC along with age, which proved highly specific and sensitive when using a novel equation—developed with logistic regression—to generate a new overall scoring system [23]. This study conveys that researchers should not only consider new variables when creating a future predictive model, but also varied patient samples and statistical analysis approaches. Regarding the variables, however, this review alone reveals how subsequent studies have either eliminated or underscored certain criteria in the LRINEC depending on their predictive relevance. Initially, there was universal criticism regarding the LRINEC’s limited variables that excluded patient age, comorbidities, and clinical presentation and placed a heavy burden on laboratory values that are not always readily available for providers upon a patient’s initial presentation [12]. However, during the development of the NECROSIS score, it was found that the absence or presence of premorbid diseases such as obesity, diabetes, and IV drug use did not have as great of an impact on the probability of developing an NSTI compared to other factors [18]. Additionally, while narrowing the model down to the three most significant variables, the NECROSIS study ruled out patient demographics like age and sex as relevant signifiers [18]. Though NECROSIS validates the LRINEC’s emphasis on laboratory values, the disregard for medical history as a potential variable, in exchange for a quick and condensed score calls for further scrutiny, as premorbid diseases such as diabetes have in fact been proven to cause NSTI in the lower extremities [24]. In fact, data from a study investigating immunocoagulopathy as an indicator for in-hospital mortality in patients with necrotizing fasciitis demonstrated a model that included the patient’s age and platelet count at admission outperformed the NLR by itself when predicting in-hospital mortality [25]. This evidence, and evidence throughout this review, suggests that a most reliable predictive model must include a broad spectrum of factors to accurately assess a patient’s risk of mortality from NSTI—it is unlikely to be accurately assessed using just one or a few metrics.

While more variables must be studied as potential signifiers for impending NSTI, one variable has remained consistent across multiple studies: white blood cell count. When validating LRINEC metrics in the NECROSIS study, WBC prevailed as a significant indicator, while other lab values were not as advantageous compared to non-laboratory signifiers [9,18]. Further, white blood cells should continue to guide future research given its proven effectiveness for both the LRINEC and NECROSIS. Researchers have in fact capitalized on this strong correlation by pursuing the significance of neutrophil, platelet, and lymphocyte levels, as these correspond to the inflammatory immune response associated with necrotizing fasciitis [15,16]. Though studies have established a positive correlation between increased NLR/PLR and mortality among those with necrotizing fasciitis, further research must be conducted for two reasons [16]. Firstly, there needs to be more evidence for their effectiveness in determining prognosis for patients with NSTI specifically, while controlling for potential comorbidities. Secondly, studies need to fine-tune and stratify cutoff values and how these translate to prognosis [16].

The POTTER score is another example of a predictive tool that was found to be less effective specifically for necrotizing soft tissue infections yet should not be neglected as a future direction of study [19]. The inclusion of POTTER in this review serves more as a stepping stone to integrating the technological concept of machine learning into clinical scoring tools. Though POTTER currently predicts postsurgical outcomes, a future application of this technology regarding NSTIs would be presurgical in deciding whether to proceed with debridement. This application can be highly beneficial especially for physicians dealing with a disease as rapidly progressive as necrotizing fasciitis. The iPhone application, which has been successful in other clinical settings, has proven to be convenient, accurate, and easy to use. However, it must be noted that as with any AI-based technology, the tool’s performance is limited to the accuracy and comprehensiveness of the data it is trained on.

Furthermore, a future, more comprehensive prognostic score should expand on the LRINEC, while taking into account three pertinent criteria based on this inclusive review. First, the model would use the more specific predictive values of the NLR and PLR to build off the proven significance of WBC, and address nuances within infectious response as opposed to a binary evaluation of elevated vs. normal WBC levels. Second, unlike the limited range of just three parameters in NECROSIS, the new score should have enough variables to address patient demographics and medical history. Third, the model would follow the example of the POTTER score by pursuing a similar interactive, nonlinear model that allows for the integration of various predictive factors on a case-by-case basis using machine learning. Machine learning takes a nonlinear approach in that the model would not assume a simple, direct relationship between a single variable and the potential development of an NSTI [26]. Instead, this application of artificial intelligence allows for computers to integrate each physician interaction into its dataset to eventually learn and use for future interchange such as the next NSTI case.

Unlike the previous studies, future research should consider using a prospective approach to validating a new score especially when machine learning is involved, as the model only improves as more data is collected. The simplicity and user adaptability of using artificial intelligence in the medical field is also only truly tested when performed in a high-stakes clinical environment. A prospective study would be most effective at a major trauma center with its own soft tissue unit and thus a high volume of patients.

In conclusion, it is imperative that clinicians further investigate the utility of the NLR and PLR in patients with NSTI through both retrospective and prospective studies, as it is evident that there is a significant lack of research on these tools that may have the potential to serve as invaluable prognostic markers for patients with possible NSTI. Once their clinical utility in NSTIs has been extensively evaluated, further research employing a machine learning algorithm modeled after the POTTER score should be the next step in designing a scoring system that encompasses all of the variables needed to evaluate and prognosticate patients with this complex condition. Ideally, this nonlinear decision tree will take into account clinical values that are readily available to the provider, such as the NLR and PLR; those hematological values in the LRINEC available in basic lab draws such as Cr, Hgb, sodium, and glucose; vital signs like SBP; and patient clinical history, such as any known premorbid conditions. This comprehensive review of the existing clinical research demonstrates that there is unlikely to be one or a few simple variables able to diagnose and prognosticate patients with NSTI. Despite this obstacle, a predictive model like the POTTER score, which effectively utilizes some of the most advanced technology in medicine today, offers the greatest potential to provide accurate evaluations of this complex and rapidly fatal condition to clinical providers in real time.

## Figures and Tables

**Table 1 jcm-14-04550-t001:** Comparing Predictive Models for NSTIs.

Predictive Score	Strengths	Limitations	NSTI Validation
LRINEC	Allows for early detection of NSTI	Heavy reliance on hematologic values; high rates of false negatives	Yes
NLR	Early marker for acute stress, simple to calculate with basic labs that are readily available	Lacks fixed cutoff values which makes interpretation difficult; does not account for confounding variables such as pre-existing comorbidities	No
PLR	Marker for inflammatory processes, simple to calculate with basic labs that are readily available	Conflicting data regarding efficacy in predicting morbidity and mortality related to NSTI	No
NECROSIS	Quick diagnostic test based on only three readily available variables; inexpensive tests required; high sensitivity (92%) for patients with one variable	Non-inclusive and limited number of variables; vague cutoff for systolic blood pressure (≤120 considered normal)	Yes
POTTER	Interactive structure allows for greater accuracy and interpretability; user-friendly cell phone application allows greater opportunity for bedside counseling of patient and family	May require information such as clinical lab values or patient medical history that is not readily available to the provider at the time of the calculation; not validated as a reliable predictor when tested among patients undergoing debridement for NSTI	No

## Data Availability

Data sharing is not applicable.

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
