# Peer review of "Predictive Models for Necrotizing Soft Tissue Infections: Are the Available Scores Trustable?"

_jcm, 2025, doi:10.3390/jcm14134550_

Round 1
Reviewer 1 Report
Comments and Suggestions for Authors
The manuscript addresses an important and clinically relevant topic: the utility and reliability of various scoring systems used to assess necrotizing soft tissue infections (NSTIs). This review fills a gap in the literature by summarizing multiple scoring models in one place and evaluating their limitations and applicability. However, several aspects of the manuscript require clarification and improvement to enhance its scientific rigor and clarity.
-
While the manuscript covers five scoring systems (LRINEC, NLR, PLR, NECROSIS, and POTTER), some are discussed more thoroughly than others. For instance, LRINEC is reviewed in depth, while PLR and POTTER feel comparatively superficial. Consider expanding the evaluation of POTTER and PLR, especially regarding limitations and potential future applications.
-
To enhance clarity and assist readers in comparing the various predictive models discussed, I strongly recommend including a summary table that outlines the key features, strengths, limitations, validation status, and diagnostic performance (e.g., sensitivity/specificity) of each scoring system (LRINEC, NLR, PLR, NECROSIS, POTTER). A well-organized comparative table would be particularly valuable for clinicians seeking to quickly assess the practical application of these tools in diagnosing or managing NSTIs. This addition would also strengthen the manuscript’s role as a reference resource.
-
The manuscript often refers to retrospective studies or single-center trials. A more critical assessment of the quality and limitations of these studies would enhance the review. For example, the PLR section mentions conflicting studies—this would be an excellent place to elaborate on potential causes of inconsistency (sample size, comorbidities, study design, etc.).
-
In some places, terms like “interactive, nonlinear model” or “machine-learning-based algorithm” are used without enough context. Consider briefly explaining what this means in lay clinical terms to benefit a broader medical audience.
-
Some references appear to be inconsistently formatted, and a few citations lack detailed context. Double-check all citations for completeness and ensure alignment with MDPI formatting guidelines.
-
The discussion section appropriately emphasizes the need for an integrated model combining various clinical and laboratory data. However, this section would benefit from more specific proposals or criteria for what such a model should include and how it could be validated.
-
The conclusion reiterates key findings but could be stronger by highlighting one or two actionable recommendations for clinicians or researchers.
Overall, this is a promising and timely review that will be valuable to clinicians and researchers working on NSTIs, but it requires revisions for clarity, completeness, and stronger critical analysis.
Author Response
Dear Ms. Xu and Reviewers,
Thank you for the opportunity to revise and resubmit our manuscript, Necrotizing Soft Tissue Infection Predictive Models: Are the available scores trustable? (jcm-3671979). We sincerely appreciate the thorough comments that will undoubtedly improve the paper’s potential clinical influence. This letter outlines the pertinent and detailed changes we have made based on each of the reviewer’s comments.
Regarding Reviewer 1:
- Comment: While the manuscript covers five scoring systems (LRINEC, NLR, PLR, NECROSIS, and POTTER), someare discussed more thoroughly than For instance, LRINEC is reviewed in depth, while PLR and POTTER feel comparatively superficial. Consider expanding the evaluation of POTTER and PLR, especially regarding limitationsand potential future applications.
- Response: To address this comment, a deeper literature review was performed for both PLR and POTTER. Unfortunately in regards to PLR, there is very minimal data available pertaining to PLR in the realm of NSTIs,thus there was little information available to add to our review To explain this to readers as to why thatsection is so much smaller than the others, the following information was added to the PLR section: “There is very little data in the current literature investigating PLR pertaining to NSTIs. The two studies discussed here are the only two found at the time of this literature review. Their methods are both retrospective in nature, however their contradictory results may be due in part to differences in sample sizes. Nonetheless, theseconflicting results and the lack of consensus throughout the current literature suggest significant further research is necessary to determine the utility of PLR in prognostication of patients with NSTI.” In regards to the POTTER section, this comment was especially helpful in pointing out an area that was lacking in pertinentexplanations and expanded evaluation. The following was added to the POTTER section to address this:“Although a strength of this decision tree-style format is that it allows the physician to follow along and interpret the questions being asked in a sequential manner, this type of format risks a significant limitation in that if it asks a question for which the provider does not have the answer readily available, there is no way forthat provider to bypass that particular question. Thus, one is to assume the decision tree is rendered useless until that value or information is obtained and submitted to the calculator. Further, as with any AI program, the application is limited to the accuracy and comprehensiveness of the dataset upon which it was trained. Anotherstrength however, is that the POTTER score has the potential to be integrated into the electronic health record(EHR), which would offer even easier usability to clinical providers [20].”
- Comment: To enhance clarity and assist readers in comparing the various predictive models discussed, I strongly recommend including a summary table that outlines the key features, strengths, limitations, validation status, anddiagnostic performance (e.g., sensitivity/specificity) of each scoring system (LRINEC, NLR, PLR, NECROSIS,POTTER). A well-organized comparative table would be particularly valuable for clinicians seeking to quickly assessthe practical application of these tools in diagnosing or managing NSTIs. This addition would also strengthen the manuscript’s role as a reference resource.
- Response: To address this extremely insightful comment, we created and added the table which can be found before the discussion section. We found this table to be highly beneficial in summarizing and clarifying thedifferences between the scores before we
discuss the generation of a new
comprehensive system in our discussion section. We also made sure to be more explicit in what this newcomprehensive score may entail and how that should be validated within our discussion and conclusion sections, which was another very helpful critique.
- Comment: The manuscript often refers to retrospective studies or single-center trials. A more critical assessment of the quality and limitations of these studies would enhance the review. For example, the PLR section mentions conflicting studies—this would be an excellent place to elaborate on potential causes of inconsistency (sample size, comorbidities, study design, etc.).
- Response: We agree with this comment and have added the following to our PLR to address this area ofopportunity: “The two studies discussed here are the only two found at the time of this literature review. Their methods are both retrospective in nature, however their contradictory results may be due in part to differences in sample sizes. Nonetheless, these conflicting results and the lack of consensus throughout the currentliterature suggest significant further research is necessary to determine the utility of PLR in prognostication of patients with NSTI.”
- Comment: In some places, terms like “interactive, nonlinear model” or “machine-learning-based algorithm” are used without enough context. Consider briefly explaining what this means in lay clinical terms to benefit a broader medical audience.
- Response: This is another incredibly insightful comment. We realized that since we had been exposed to the terms machine-learning-based algorithm and nonlinear model throughout our literature review, we failed to identify these as terms we should explain to our own readers. To address this, we have added the following to the POTTER section: “The interactive calculator utilizes a machine-learning based algorithm, an artificialintelligence (AI) application in which machines recognize patterns based on their experiences from a given dataset, which allows for nonlinear variability as well as higher accuracy and interpretability than classicaldecision-tree methods [20]. Essentially, the use of the machine-learning method called Optimal Classification Trees (OCT) allows the decision-tree to adapt to each input and re-calculate risk after each variable. Further, the variables evaluated at each level of the tree are not the same, meaning the questions asked at each level change depending on the response to the previous question. This function is what allows for the nonlinear interactions between the variables and ultimately increases the accuracy of the tree, in contrast to the fixed interactions utilized in a classical logistic “ Additionally, the following was added to our discussion section to provide further clarification: “Machine learning takes a nonlinear approach in that themodel would not assume a simple, direct relationship between a single variable and the potential development of an NSTI [25].”
- Comment: Some references appear to be inconsistently formatted, and a few citations lack detailed context. Double-check all citations for completeness and ensure alignment with MDPI formatting
- Response: Thank you for identifying these errors for We have thoroughly reviewed our references list andhave ensured all are aligned with MDPI formatting guidelines. We also reviewed our in-text citations and haveadded context where it was lacking, specifically in places where the authors of a study were identified (example: “ et al”) and then the study discussed.
- Comment: The discussion section
appropriately emphasizes the need for an integrated model combining various clinical and laboratory data. However, this section would benefit from more specific proposals or criteria for what such a model should include and how it could be validated.
- Response: We agree. To address this, we added explicit examples of what this new comprehensive score may entail and how it should be validated. Specifically, we outlined three criteria that the new score should include (NLR/PLR, medical history, and machine learning) and suggested that a prospective study testing the newscore should be conducted at a major trauma center, as opposed to a retrospective study. The following was added to the discussion section to address this: “Furthermore, a future more comprehensive prognostic score should expand on LRINEC, while taking into account three pertinent criteria based on this inclusive review. First, the model would use the more specific predictive values of NLR and PLR to build off the proven significance of WBC, and address nuances within infectious response as opposed to a binary evaluation of elevated vs normal WBC levels. Second, unlike the limited range of just three parameters in NECROSIS, thenew score should have enough variables to address patient demographics and medical Third, the modelwould follow the example of the POTTER score by pursuing a similar interactive, nonlinear model that allows integration of the various predictive factors on a case-by-case basis using machine learning. Machine learningtakes a nonlinear approach in that the model would not assume a simple, direct relationship between a single variable and the potential development of an NSTI [25].”
- Comment: The conclusion reiterates key findings but could be stronger by highlighting one or two actionable recommendations for clinicians or researchers.
- Response: This was another very insightful comment. To address this, we added the following paragraph to provide a stronger conclusion with actionable items for further research: “In conclusion, it is imperative that clinicians further investigate the utility of NLR and PLR in patients with NSTI through both retrospective and prospective studies, as it is evident that there is a significant lack of research in these tools that may have thepotential to serve as invaluable prognostic markers for patients with possible NSTI. Once their clinical utilityin NSTIs has been extensively evaluated, further research employing a machine-learning algorithm modeled after the POTTER score should be the next step in designing a scoring system that encompasses all of the variables needed to evaluate and prognosticate patients with this complex condition. Ideally, this nonlinear decision-tree will take into account clinical values that are readily available to the provider, such as NLR andPLR, those hematological values in LRINEC available in basic lab draws such as Cr, Hgb, sodium and glucose, vital signs like SBP, as well as patient clinical history such as any known premorbid conditions. This comprehensive review of existing clinical research demonstrates that there is unlikely to be one or a few simplevariables able to diagnose and prognosticate patients with NSTI. Despite this obstacle, a predictive model like the POTTER score, which effectively utilizes some of the most advanced technology in medicine today, offers the greatest potential in providing accurate evaluations of this complex and rapidly fatal condition to clinical providers in real-time.”

Reviewer 2 Report
Comments and Suggestions for Authors
This minireview discusses several systems for assessing necrotizing fasciitis and necrotizing soft tissue infections. Overall, the information presented is helpful, though there are a few weaknesses in the presentation.
Major points
1. A better distinction between necrotizing fasciitis and NSTIs in general is needed. A brief discussion of the causative agents would also help orient the reader.
2. The Potter score appears to be for post-surgery, yet the rest of the tests are to help make the decision to proceed to surgery or not.
3. Systolic blood pressure <120 is normal. Unclear how this fits into Necrosis score.
4. The claim: "absence or presence of premorbid diseases such as obesity, diabetes, and IV drug use did not impact the probability of developing an NSTI" is incorrect. Diabetes is associated with NSTIs, especially in the toes.
Minor Points
1. Statements with "Author et al" need to be referenced
2. Modifications to LRINSEC, like J-LRINSEC (PMID: 39916636) should be considered.
Author Response
Regarding Reviewer 2:
- Comment: A better distinction between
necrotizing fasciitis and NSTIs in general is needed. A brief discussion of the causative agents would also help orient the reader.
- Response: Thank you for identifying this insufficiency in our introduction. We have addressed this byclarifying necrotizing fasciitis as a type of necrotizing soft tissue infection along with other types of NSTI and broadened the focus of our article to NSTI overall as opposed to necrotizing fasciitis in the Weadded the following information in our introduction section: “Also known as the “flesh eating disease,”necrotizing soft tissue infections (NSTI) such as necrotizing fasciitis, necrotizing myositis, and necrotizingcellulitis are categorized as bacterial infections of the subcutaneous soft tissue surrounding muscles and nervesthat can ultimately result in death depending on severity and timeliness of treatment [1]. NSTIs are most often caused by group A β-hemolytic streptococci, staphylococcal species, or both in combination following open wounds such as trauma or surgery [2].”
- Comment: The Potter score appears to be for post-surgery, yet the rest of the tests are to help make the decision to proceed to surgery or not.
- Response: Thank you for identifying this. We have addressed this concern by explaining the utility of including the POTTER score in our review to serve as a potential model for the generation of a future NSTI score in terms of determining whether to proceed with debridement, not necessarily as a direct score to be used for NSTIs. We have added the following to the POTTER section: “The inclusion of POTTER in this review serves more as a stepping stone to integrating the technological concept of machine-learning intoclinical scoring Though POTTER currently predicts post-surgical outcomes, a future application of thistechnology regarding NSTI’s would be pre-surgical in deciding whether to proceed with debridement.”
- Comment: Systolic blood pressure <120 is Unclear how this fits into Necrosis score.
- Response: This is a valuable comment and is very important that we address in our In our discussion section, we have now acknowledged that the NECROSIS score disregarded several significant factors for the sake of creating a condensed and efficient model that may actually be too restricted. We also acknowledged the unusually stable cutoff of SBP≤120 in the NECROSIS score by looking back at the literature, which clarified that the authors came to this cutoff based on their sample of patients with NSTI and without NSTI. We included how they understood this value as a chance to advise physicians to be wary of evenhemodynamically stable patients. The following information was added to the NECROSIS score section: “The study found that among all vital signs, systolic blood pressure still differed the most between patients with andwithout necrotizing infection, with an unusually stable cutoff of 120 mmHg. The authors use this value tocaution physicians to be wary of potential NSTI even when patients appear hemodynamically stable, as thestudy found that only 1 in 5 NSTI patients met criteria for septic shock upon initial presentation.” Further, the following section regarding the importance of including the patient’s medical history for greater accuracy in an NSTI scoring system was moved up to provide a critique of the NECROSIS score, which may be too simplifying in its inclusion of only three variables: “In fact, data from a study investigatingimmunocoagulopathy as an indicator for in-hospital mortality in patients with necrotizing fasciitis demonstrateda model that included the patient’s age and platelet count at admission outperformed NLR by itself when predicting in-hospital mortality [24]. This
evidence and evidence throughout
this review suggests that a most reliable predictive model must include a broad spectrum of factors to accurately assess a patient’s risk of mortality from NSTI - it is unlikely to be accurately assessed using just one or a few metrics.”
- Comment: The claim: "absence or presence of premorbid diseases such as obesity, diabetes, and IV drug use did not impact the probability of developing an NSTI" is incorrect. Diabetes is associated with NSTIs, especially in the toes.
- Response: Thank you for identifying this misleading information in our paper. We agree with your evaluation and have revised our content to align with the current literature. The following was added to the NECROSISscore section: “However, during the development of the NECROSIS score, it was found that the absence or presence of premorbid diseases such as obesity, diabetes, and IV drug use did not have as great of an impact on the probability of developing an NSTI compared to other factors [18]. Additionally, while narrowing the model down to the three most significant variables, the NECROSIS study ruled out patient demographics like age and sex as relevant signifiers [18]. Though NECROSIS validates LRINEC’s emphasis on laboratoryvalues, the disregard for medical history as a potential variable, in exchange for a quick and condensed scorecalls for further scrutiny, as premorbid diseases such as diabetes have in fact been proven to cause NSTI in the lower extremities [23].”
- Comment: Statements with "Author et al" need to be referenced
- Response: Thank you for identifying these areas - this was corrected throughout the review and extra context was provided in addition to adding references to those sentences.
- Comment: Modifications to LRINSEC, like J-LRINSEC (PMID: 39916636) should be
- Response: We appreciate you highlighting an opportunity for further discussion on future directions based on ongoing modifications of LRINEC, especially given the recent publication of the J-LRINEC score in 2025. We addressed J-LRINEC early in the discussion when acknowledging how the prognostic value of LRINECcan be improved in several ways in addition to selecting new variables, as J-LRINEC looks at a specific patient population, and uses a different statistical model in creating their final predictive calculation: “In fact, a newly published score titled J-LRINEC tests the validity among Japanese patients using all the same variables as LRINEC along with age, which proved highly specific and sensitive when using a novel equation–developed with logistic regression–to generate a new overall scoring system [23]. This study conveys that researchers should not only consider new variables when creating a future predictive model, but also varied patient samples and statistical analysis approaches.”
We find that after addressing all of the reviewer
comments and making necessary alterations, our paper has become a much stronger candidate for publication.
We look forward to hearing from you soon regarding our revised manuscript. Thank you for your consideration of this revised manuscript.
Sincerely,
Ms. Sophie Tran, Ms. Kerry Pullano, Dr. Marcelo AF Ribeiro, Dr. Sharon Henry

Round 2
Reviewer 1 Report
Comments and Suggestions for Authors
The authors have clearly made a significant effort to strengthen their review, integrating feedback effectively and enhancing the manuscript’s clarity, depth, and relevance. Therefore, I recommend acceptance in present form.